## Replications

cognition

kinaesthetic memory, working memory, motor memory

**Author for correspondence:**
Elena Nicoladis
e-mail: elenan@ualberta.ca

# Towards a reliable measure of motor working memory: revisiting Wu and Coulson's (2014) movement span task

Elena Nicoladis and Rielle Gagnon

University of Alberta, Edmonton, Alberta Canada, T6G 2R3

EN, 0000-0002-5065-9159

Some researchers have argued that motor working memory is relatively independent from visuospatial working memory and underlies the learning and processing of motor tasks, like gesture comprehension. To allow systematic testing of these claims, Wu & Coulson 2014 *Psychol. Sci.* **26**, 1717–1727. (doi:10.1177/0956797615597671) proposed a novel measure of motor working memory, the movement span task. Some studies have reported that the movement span task has a high degree of validity. The purpose of the present study was to attempt to replicate Wu & Coulson 2014 *Psychol. Sci.* **26**, 1717–1727. (doi:10.1177/0956797615597671) in the following ways: (1) the high correlation between movement span and movement recall scores and (2) the lack of correlation between the movement span task on the one hand and visuospatial and verbal working memory on the other. In the present study, we found a high correlation between the movement span and recall scores as well as most measures of visuospatial memory. However, the size of these correlations was similar to that reported by Wu and Coulson, suggesting that the significance may be related to sample size. In other words, motor working memory may be weakly related to visuospatial memory. By contrast, there were weak correlations between the movement span task and verbal memory. In sum, we found the same pattern of results observed by Wu & Coulson 2014, 1717–1727. (doi:10.1177/0956797615597671).

## 1. Introduction

Working memory refers to the capacity to store unrelated units of information for a short period of time ([1,2]; cf. [3]). There is some evidence that working memory differs by modality, with relatively independent auditory (or verbal) and visuospatial stores [1,2,4–15]. Working memory capacity within a modality

can be related to the learning and processing of modality-specific information. For example, verbal memory capacity has been implicated in both language production [16] and language comprehension [17]. Similarly, visuospatial memory has been linked to processing in spatial tasks, such as situational awareness while driving [18] and learning novel routes [19].

Learning and processing motor tasks could involve memory for bodily movements [20,21]. Some researchers have argued that motor working memory is a subset of visuospatial memory [4,5,22–26]. For example, Seidler et al. [24] showed that visuospatial memory was correlated with the rate of motor learning. By contrast, other researchers have argued that motor working memory is relatively independent from other working memory stores [21,27–32]. For example, Smyth & Pendleton [30] found that there was interference from body movements on memory for body movements, but less interference from saying words or watching spatial targets. Proponents of the view that motor memory is somewhat independent from visuospatial memory usually distinguish motor memory, memory for explicit, unrelated, and novel movements [21] from procedural memory, the memory for implicit, sequenced, and habitual movements [33,34]. If there were an independent motor working memory, it should be related to learning and processing motor information, like co-speech gestures [21].

In order to test these arguments, it is essential to have a valid and reliable measure of motor working memory. Wu & Coulson [21] proposed such a measure. Wu & Coulson's [21] movement span task was designed to assess memory for body movements by having participants repeat unfamiliar, unconventional gestures. Wu & Coulson [21] found high test-retest reliability for the movement span task after one week among twelve participants. By analogy with other working memory tasks, Wu & Coulson [21] calculated two scores for participants: the span score and the total recall score. The span score refers to the highest number of gestures accurately recalled by participants. The total recall score refers to the total accuracy points a participant accrued across all trials. They reasoned that the span and the total scores would be highly related. As they predicted, there was a correlation of 0.80 between the two scores. Finally, Wu and Coulson [21] found that performance on the movement span task was unrelated to verbal memory and to visuospatial memory, as measured by a Corsi block task. This result suggests that performance on this task is relatively independent from verbal and visuospatial memory.

To verify the validity of the movement span task, one important approach is to test if performance on this task is related to learning or processing motor-related tasks. Indeed, a number of studies have shown that performance on the movement span task is related to processing or learning co-speech gestures or signs in signed languages. For example, Wu & Coulson [21] showed that the movement span task predicted participants' performance on a gesture classification task. In this task, participants were asked to identify whether a co-speech gesture was congruent or incongruent in meaning with the accompanying speech. Similarly, Wu & Coulson [35] found that movement span performance predicted how much people relied on gestures in their understanding of discourse. Finally, Martinez & Singleton [36] showed that performance on this task was a significant predictor of hearing people's success in learning novel signs.

The purpose of this study was to replicate Wu & Coulson's [21] movement span task to verify the reliability. Wu & Coulson's [21] study included only 90 participants, a relatively small sample size for a psychometric study. In this study, we included just more than double the number of participants. We expected to replicate two of their findings: (1) a high correlation between the span and total scores and (2) no relationship between the movement span task and either visuospatial memory or verbal memory. To perform this replication study, we use data from a previously published study [37]. The analyses presented here do not duplicate any of those in the published study. To test the second point, we included two measures of visuospatial memory: forward and backward Corsi blocks. Some research has found no difference between forward and backward Corsi block spans [38]. One study showed that backward Corsi block span might be more sensitive in identifying visuospatial ability [39].

## 2. Method

Our study was part of a larger study in which participants completed a battery of seven working memory tasks. The study methods were approved by the Research Ethics Board of the University of Alberta prior to recruiting participants. We present the data here for only the motor memory task, the visuospatial memory tasks, and the verbal memory task. One hundred and eighty-five people participated, about half were musicians (cf. [21], who included 90 participants and did not enquire about their participants' musical background). Five other people participated in the study but did not

complete both the motor and visuospatial memory tasks so their data were not included in the analyses of this study. Almost all participants were recruited via the psychology research participation pool and received partial course credit for completion of the study, however, snowball sampling was also used to recruit several additional participants. These additional participants received a $10.00 honorarium for their participation. Participants were provided with full information about the study's purpose and procedures before signing to indicate that they were participating with informed consent. All participants were either university students or were previously university students and had graduated university within a year prior to their participation. Participants were between the ages of 17 and 29 ($M = 19.61$, s.d. = 1.98). Wu & Coulson [21] did not report the age of their participants. However, since they recruited university students and offered course credit for participation, it is likely that the ages of the participants are similar.

Our previous analyses showed that the musicians scored significantly higher than the non-musicians on the forward visuospatial memory task but not on the motor memory task [37]. The effect size for the visuospatial memory difference was small ($d = 0.300$). In the present analyses, we do not distinguish the musicians and non-musicians. In the discussion, we will return to the possible effects of musical training on the results.

## 2.1. Motor memory task

The same 45 gestures included in Wu & Coulson's [21] movement span task were included in the present study. Following Wu & Coulson [21], the movement span consists of five levels, with three trials of the same difficulty per level. Each participant completed all five levels of the task. For each trial, participants watched a short video on a computer screen. The videos were of a camera facing person who performed a series of unconventional gestures. Participants were then cued with an auditory signal when the video was over, and were instructed to recreate the gestures they had seen on the screen by mirroring the person in the video (i.e. if the person on the video used their left hand, participants were supposed to use their right hand). All participants did a practice trial and received feedback from the experimenter. All three trials in level one contained one movement, and as the level increased by one, so did the number of movements in the level, so that in level five, trials contained five movements.

Following Wu & Coulson [21], participants were awarded one point for each correct movement they reproduced, and half a point for partially correct movements. For a movement to be considered partially correct, participants could only be missing one parameter of the movement, such as incorrect hand orientation, trajectory, or handshape (following the catalogue of relevant movement parameters from [21]). If more than one parameter of the movement was incorrect, no points were awarded. Also following Wu & Coulson [21], the order of the movements was not taken into account in the scoring. The span score was calculated by awarding one point for each level at which participants achieved at least half of the available points, for a maximum span score of 5. If participants achieved less than half of the available points at any given level, but then achieved at least half of the available points for a subsequent level, a half-point was awarded for those levels. The total score was calculated by adding all the awarded points across all trials for each level, for a maximum total score of 45. Higher scores indicated greater motor memory capacity. Our method of coding was more stringent than that of Wu & Coulson [21]. In their study, participants needed to recall correctly only a single trial at any given level to advance to the next. For that reason, our scores may be lower than theirs.

Participants were videotaped while performing this task for scoring accuracy and interrater reliability. The percentage of total correct movements was calculated to account for missing trials due to camera malfunction and technical difficulties. Two independent raters coded the movement span videos. Each rater also coded two videos that had been coded by the other rater. Wu & Coulson [21] repeated high inter-coder agreement, with correlations ranging between 0.80 and 0.92 for the span score and between 0.84 and 0.98 for the total score. In our study, the two raters had very high and significant correlations in their rating of the total scores, $r_2 = 0.989$, $p < 0.02$. Total score differences between the videos coded by both raters were no greater than one point. The mean total score rating for Coder 1 was 15.8 and for Coder 2 15.6. The two raters agreed perfectly on the span score.

## 2.2. Visuospatial memory tasks

To determine whether the movement span task was measuring a distinct body/motor component of memory, rather than some form of visuospatial memory, participants also completed the forward and backward Corsi block tapping task as measures of visuospatial memory and visuospatial working

**Table 1.** Averages (SDs) working memory scores.

|  | present study | Wu & Coulson [21][a] |
|---|---|---|
| movement span: total | 14.1 (4.6) | 23.8 (5.7) |
| movement span: span | 1.8 (0.7) | 3 (1) |
| forward Corsi: total | 9.2 (1.8) | 20.4 (4.6) |
| forward Corsi: span | 5.6 (0.9) | 7 (1) |
| backward Corsi: total | 7.3 (2.0) | n.a. |
| backward Corsi: span | 4.7 (1.1) | n.a. |
| forward digit span | 5.6 (1.2) | n.a. |

[a]Data are from Wu & Coulson [21] p. 7, table 1.

memory, respectively. In the forward Corsi block tapping task, participants saw a series of blocks highlighted in a sequence on a computer screen. Participants then had to tap the blocks back in the same order they saw them highlighted by touching the screen. There were two trials consisting of sequences of the same length in each level of the task, beginning with a sequence length of two, and working up to a sequence length of nine. In this way, the task gets progressively more difficult as the sequence length increases at higher levels. This task has a discontinue rule, meaning participants must correctly tap both trials in each level to progress to the next level. If participants tapped the first trial in a level incorrectly, the second trial was still completed, but they did not move onto the next level. The total score, namely the number of correct trials across all levels, was calculated for every participant (theoretical range: 0–45). The span score was the number of levels at which participants received at least half of the available points (theoretical range: 0–5). Half points denote occasions when participants received fewer than half of the available points at one level, but then received at least half of the available points on a subsequent level.

The same procedure was used for the backward Corsi block tasks, except participants had to tap the blocks back in the reverse order that they appeared. Both tasks were marked on an all-or-nothing basis, meaning participants only received a point for the trial if all blocks were tapped correctly. Only one version of the task was used, so all participants saw the same block sequences. For the visuospatial memory score tasks, the total score was calculated as the total number of correct responses and the span score was the highest number of items remembered without error.

## 2.3. Verbal memory task

To assess verbal memory, we used the forward digit span task. The researcher read a list of digits out loud to participants. The number of digits increased with each trial, starting with two digits and going to nine digits. Every participant completed all list lengths. This task was scored so that if participants said any incorrect digit in the list, they were counted as zero. Their score on the forward digit span task was the highest list size that they attained with no errors. Thus, for this task, the span and the total scores were equivalent. Scores on this tasks could theoretically range from 0 to 9. In our study, the participants' scores ranged from 2 to 8.

## 3. Results

Table 1 summarizes the descriptive statistics for our study in comparison to that of Wu & Coulson [21]. Based on one-sample $t$-tests, Wu & Coulson's participants had higher total recall scores, $t_{184} = -28.58$, $p = 0.00$, and higher span scores, $t_{184} = -22.76$, $p = 0.00$, than our participants did. Our participants also scored significantly lower on the forward Corsi block task, both on span scores, $t_{183} = -19.85$, $p = 0.0001$, and total scores, $t_{183} = -83.57$, $p < 0.0001$.

Wu & Coulson [21] argued that movement span and movement recall captured largely the same construct because they found a high correlation between the two ($r = 0.80$). We also examined the correlation between the two scores after standardizing the data to eliminate differences in scale size using $z$-scores, following Wu & Coulson [21]. As can be seen in table 1, we, too, found a significant correlation between the span and total scores, $r_{184} = 0.724$, $p < 0.001$.

**Table 2.** Correlations between memory tasks.

|  | 1 | 2 | 3 | 4 | 5 | 6 |
|---|---|---|---|---|---|---|
| 1. movement span: total score | — | | | | | |
| 2. movement span: span score | 0.724** | — | | | | |
| 3. forward Corsi block (total score) | 0.254** | 0.227** | — | | | |
| 4. forward Corsi block (span score) | 0.284** | 0.236** | 0.967** | — | | |
| 5. backward Corsi block (total score) | 0.212** | 0.134 | 0.339** | 0.345** | — | |
| 6. backward Corsi block (span score) | 0.188* | 0.129 | 0.303** | 0.304** | 0.969** | — |
| 7. forward digit span | 0.110 | 0.120 | 0.170* | 0.163* | 0.033 | 0.026 |

$**p < 0.01$ $*p < 0.05$.

Finally, we examined whether the movement span task was correlated with measures of visuospatial and verbal memory. Recall that Wu & Coulson [21] found no significant correlations between forward Corsi block span and movement span ($r = 0.13$) and movement recall ($r = 0.22$). The correlations for our study are summarized in table 2. Critically, the correlations between forward Corsi block span and movement span ($r = 0.25$) and movement recall ($r = 0.23$) were similar in size to those found in Wu & Coulson [21]. In our study, however, they were statistically significant because of our larger sample size. Wu & Coulson [21] used different measures of verbal memory than we did, but the correlations ranged from 0.09 to 0.18 to movement span performance. Again, the magnitude of correlations in our study were very similar.

## 4. Discussion

Our results suggest that many aspects of Wu & Coulson [21] were replicated. Our scores on the movement span task were significantly lower than those reported by Wu and Coulson [21]. We used a more stringent coding scheme for accuracy, which is the most likely reason for lower scores. However, it is also possible that our study included participants with lower memory abilities all around. In support of that possibility, our participants also scored lower than the participants in Wu & Coulson [21] on the visuospatial memory task. Another possibility is that our participants were fatigued because we administered seven different working memory tasks within an hour, while Wu & Coulson [21] asked their participants to do a more varied set of tasks.

In any case, like Wu & Coulson [21], we found a high correlation between the span and recall scores. It should be noted that for many working memory tasks, the span and total scores are often co-dependent in their method of calculation. Thus, it is not surprising that the correlation between span and total scores on the movement span task, forward Corsi blocks task, and backward Corsi blocks task are so high (table 2) that for most research, it would probably be sufficient to include only one (as suggested by [21]).

Finally, like Wu & Coulson [21], we found low correlations between the movement span task and verbal memory. However, while Wu & Coulson [21] found no significant correlations between visuospatial memory (forward Corsi blocks) and movement span, we did. However, it is important to keep in mind that the magnitude of the correlations reported in both studies was similar and our study included about twice as many participants as the Wu & Coulson [21] study. These results are therefore suggestive that there is a small relationship between visuospatial and motor memory, such that a large number of participants is required to see a significant correlation.

In the present study, we also included another measure of visuospatial working memory, backward Corsi blocks. Some researchers have argued that the backward Corsi blocks task is a better measure of visuospatial memory than the forward Corsi blocks task [39]. If so, as can be seen in table 2, the correlations between the movement span task and backward Corsi blocks were also modest, even when significant. These results suggest motor working memory is neither simply a subset of visuospatial memory (as claimed by [4]) nor entirely independent of visuospatial memory (as claimed by [21]). Instead, motor working memory seems to be weakly related to visuospatial memory. One possible reason for this weak relationship is that motor working memory may work with visuospatial working memory in the practice of carrying out motor tasks (see [40]). However, some visuospatial

tasks may be carried out with only minimal motor memory. Future research can focus on how memory for visual, spatial, and motor stimuli is related in real-life tasks. This research could also include other measures of visuospatial memory. Some studies have shown that the Corsi blocks task is not a pure measure of visuospatial memory [23,41,42]. Thus, the inclusion of other visuospatial tasks could lead to a more refined understanding of the function of motor memory relative to memory in other modalities.

One other study reported a high correlation ($r = 0.391$) between movement memory and visuospatial memory [36]. It is not entirely clear why such a high correlation was found relative to the present study and that of Wu & Coulson [21]. One possibility is that Martinez & Singleton [36] coded accuracy on the visuospatial memory task by awarding partial points for partial sequences, as long as they were recalled in the presented order. By contrast, we used an all-or-none coding method for each sequence length. This coding approach used in Martinez & Singleton [36] could have been closer to the coding used for the movement span task [21,36], allowing for partially correct memory for movements. Future studies can test that interpretation by comparing the results using different coding approaches for accuracy.

In designing these studies, it is important to keep in mind that experience can change the structure of working memory. For example, musical training can lead to a high degree of integration of motor and auditory memory ([43]; see review in [44]). We have shown that for musicians, performance on the movement span task was relatively independent of visuospatial memory, verbal memory and memory for their own movements [37]. By contrast, for non-musicians, performance on the movement span task was related to one measure of verbal memory, visuospatial memory, and one measure of memory for their own movements [37]. These results suggest that some experiences can fundamentally change how integrated motor memory is with other memory modalities. Future studies could also include measures of experiences known to impact the structure of working memory, such as musical training.

In conclusion, we have replicated many key aspects of Wu & Coulson's [21] study, contributing to the evidence that their movement span task is a reliable measure. With a reliable measure of motor memory, future research can better address questions of how motor memory relates to other aspects of memory.

Data accessibility. URL to protocol: https://osf.io/gw2rx. This article received results-blind in-principle acceptance (IPA) at Royal Society Open Science. Following IPA, the accepted Stage 1 version of the manuscript, not including results and discussion, was preregistered on the OSF (URL). This preregistration was performed after data analysis.

Authors' contributions. E.N. drafted the manuscript and did some of the data analysis. R.G. collected the data, participated in data analysis and critically revised the manuscript. Both authors gave final approval for publication and agree to be held accountable for the work performed therein.

Competing interests. The authors declare no competing interests.

Funding. This study received funding from a Discovery grant (no. 239851) to the first author from the Natural Sciences and Engineering Research Council of Canada.

Acknowledgements. The authors thank all of the research assistants who made this study possible by running participants and coding data: Maria Chow, Mariam Dar, Sunny Dhonkal, Beverly Michel Baluyot, Gurishar Dami, Mathew Gorman, Nicole Love, Anna Kuc, Habeebah Mohammad, Zaahidah Ali and Tehseenah Zahrah.

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
