## [Reviewer comments · Royal Society Open Science]

Review History

RSOS-200237.R0 (Original submission)

Review form: Reviewer 1 (David Martinez)

Do you have any ethical concerns with this paper?

No

Have you any concerns about statistical analyses in this paper?

No

Recommendation?

Accept with minor revision

Comments to the Author(s)

The study described herein aims to replicate Wu and Coulson's (2014) results, suggesting that immediate memory for body movements is unrelated or, at best, weakly related, to immediate visuospatial memory.

I believe the manuscript describes a valid and robust replication attempt however, I think we need a bit more detail so that readers can judge how close of a replication this is. On page 5 of the

manuscript, lines 26-31, it is not clear what is meant by a "component." Moreover, an example of incorrect hand orientation is provided but, how accurate must one be? Is being off by 15 degrees acceptable? What about 45 degrees? I believe the instructions provided to raters and/or examples should be provided in an appendix.

It is also unclear whether participants were asked to recall body movements in serial order or any order---I assume free recall but this wasn't explicitly stated.

I have just one more suggestion for the current manuscript. Wu and Coulson (2014) also provided measures of immediate verbal memory. Although it is somewhat out of the scope of this study, it would be helpful to include a verbal measure if it was part of the battery of tasks given in the larger study. If you find that the Corsi task correlates with the movement span task, then it may be because of domain-specific processes and/or domain-general processes--- including a verbal measure would aid in adjudicating between these possibilities.

Review form: Reviewer 2

Do you have any ethical concerns with this paper?

No

Have you any concerns about statistical analyses in this paper?

Yes

Recommendation?

Accept with minor revision

Comments to the Author(s)

The question of whether motor span and the associated construct of kinaesthetic working memory (KWM) can be dissociated from visuospatial working memory (VSWM) as assessed by the Corsi Block task (which involves visual encoding and the generation of motor sequence to signal recall) is an important issue with potential relevance for the study of communicative movements such as co-speech gestures, signs in signed languages, and dance. The conclusion in Coulson and Wu (2014) that the two skills are dissociable was based on a study of 90 participants, a relatively low number for a psychometric study. Thus, replication of the observed low correlation coefficients (-0.01 with Corsi Span and 0.07 with Corsi Total) with more than double the original number of participants (viz. 200) would bolster this aspect of the original study. Likewise, the observation of reliable correlations between scores for movement span and Corsi span would suggest the original result was a Type II error and would represent a genuine contribution to the literature.

The main way that the present study differs from the original is in the details of the Corsi Block Span test. In Wu and Coulson (2014), levels ranged from four to nine (c.f. in the present study, levels ranged from two to nine), and five trials (viz. sequences) were presented at each level (c.f. two trials were presented at each level in the present study). In Wu and Coulson (2014), participants only needed to correctly recall a single trial at a given level in order to advance to the next. In the present study, participants needed to correctly both trials at a given level in order to advance to the next level. In both studies, the total score was computed by summing all trials in which the participant correctly recalled all of the items in the sequence. However, the subtle differences between the administration of the test would predictably lead to differences in the total score values. Likewise, subtle differences between the way span scores were computed two studies might also lead to differences. In Wu and Coulson (2014), the span score was the highest level a participant completed (and because completion only required a single correct trial, there were no half points awarded). In the present study, participants were awarded a half-point to

their span score if they advanced to a level and only had one correct trial. So, it is in principle possible that a failure to replicate might be attributable to these differences. However, given that there are a number of variants of the Corsi Block task that have been usefully deployed as a measure of visuospatial ability, these differences are unlikely to undermine the conclusions.

Moreover, another way that the present study differs from Wu and Coulson (2014), is that the authors here have employed two variants of the Corsi Block task – forward and backward. Although the two versions arguably assess slightly different cognitive abilities, with the backward span being more sensitive to the deployment of executive resources to track order (Cornoldi & Mammarella, 2008; Vandierendonck & Szmalec, 2003), they tend to be highly correlated with one another. The present study thus affords the possibility of conducting an even more robust assessment of the hypothesis that kinaesthetic and visuospatial WM abilities are dissociable from one another than the original study. This would involve both assessing the intercorrelation (or consistency via Cronbach’s alpha) of each participant’s forward and backward span scores in order to demonstrate their construct validity, and the separate correlation of each of the two Corsi span scores with participants’ span scores on the motor working memory task.

Cornoldi, C., & Mammarella, I. C. (2008). A comparison of backward and forward spatial spans. *The Quarterly Journal of Experimental Psychology*, 61(5), 674-682.

Vandierendonck, A., & Szmalec, A. (2003). An asymmetry in the visuo-spatial demands of forward and backward recall in the Corsi blocks task. *Imagination, Cognition and Personality*, 23(2), 225-231.

Review form: Reviewer 3

Do you have any ethical concerns with this paper?

No

Have you any concerns about statistical analyses in this paper?

No

Recommendation?

Accept with minor revision

Comments to the Author(s)

In this replication study, the authors examine the validity of a recently published movement span task. Given the relatively limited sample size reported in the original study, I think the need for a replication is clear.

The method and results outlined here appear to be a faithful replication of the original paper. Some suggestions to improve the paper:

- I would like to see more details in the introduction on how the movement span task differs from previous attempts to determine the separability of movement from other WM subsystems (e.g., visuospatial WM). Why is the current approach showing a separate WM system while other papers have not? Similarly I'd like to see some further explanation on the rationale behind looking at span scores and total accuracy. These are obviously related because span scores depend on accuracy. Why is it meaningful to look at both scores? I understand the need to replicate the approach from the original paper, but it wasn't clear to me why these are the best DVs.

- Also in the intro, the authors can do more to build the case for why a replication is needed. While I can infer that from the method, a dedicated section in the intro laying out the need would strengthen the rationale.

- Once the manuscript is complete with data and a discussion, I encourage the authors to discuss the limitations of the Corsi blocks task in measuring visuospatial WM and consider including at least one other visuospatial span task (e.g., symmetry or rotation span tasks) if data is not yet collected. Future work in this area should expand the visuospatial measures used to validate the movement span task to ensure it is actually separate from visuospatial WM.

Decision letter (RSOS-200237.R0)

25-Mar-2020

Dear Dr Nicoladis

On behalf of the Editors, I am pleased to inform you that your Manuscript RSOS-200237 entitled "Toward a reliable measure of motor working memory: Revisiting Wu and Coulson's (2014) movement span task" deemed suitable for in-principle acceptance in Royal Society Open Science subject to minor revision in accordance with the referee and editor suggestions. Please find their comments at the end of this email.

The reviewers and handling editors have recommended publication, but also suggest some minor revisions to your manuscript. Therefore, I invite you to respond to the comments and revise your manuscript.

Please you submit the revised version of your manuscript within 4 weeks (i.e. by the 02-Apr-2020). If you do not think you will be able to meet this date please let me know immediately.

When submitting your revised manuscript, you will be able to respond to the comments made by the referees and upload a file "Response to Referees" in the "File Upload" step. You can use this to document any changes you make to the original manuscript. In order to expedite the processing of the revised manuscript, please be as specific as possible in your response to the referees.

Full author guidelines can be found here <https://royalsocietypublishing.org/rsos/replication-studies#AuthorsGuidance>.

Kind regards,
Professor Chris Chambers
Royal Society Open Science
openscience@royalsociety.org

on behalf of Professor Chris Chambers (Associate Editor) and Chris Chambers (Registered Reports Editor, Royal Society Open Science)
openscience@royalsociety.org

Associate Editor Comments to Author (Professor Chris Chambers):

Associate Editor: 1

Comments to the Author:

Three expert reviewers have now assessed the manuscript. All are positive overall, while also noting areas for minor revision, chiefly in explaining and justifying deviations from the original methodology. In relation to one point from Reviewer 3 (prompting the authors to "build the case for why a replication is needed") please note that this is not a criterion by which replication studies are assessed at RSOS, therefore the authors need not respond to this point in their revised manuscript.

Reviewers' comments to Author:

Reviewer: 1

Comments to the Author(s)

The study described herein aims to replicate Wu and Coulson's (2014) results, suggesting that immediate memory for body movements is unrelated or, at best, weakly related, to immediate visuospatial memory.

I believe the manuscript describes a valid and robust replication attempt however, I think we need a bit more detail so that readers can judge how close of a replication this is. On page 5 of the manuscript, lines 26-31, it is not clear what is meant by a "component." Moreover, an example of incorrect hand orientation is provided but, how accurate must one be? Is being off by 15 degrees acceptable? What about 45 degrees? I believe the instructions provided to raters and/or examples should be provided in an appendix.

It is also unclear whether participants were asked to recall body movements in serial order or any order---I assume free recall but this wasn't explicitly stated.

I have just one more suggestion for the current manuscript. Wu and Coulson (2014) also provided measures of immediate verbal memory. Although it is somewhat out of the scope of this study, it would be helpful to include a verbal measure if it was part of the battery of tasks given in the larger study. If you find that the Corsi task correlates with the movement span task, then it may be because of domain-specific processes and/or domain-general processes--- including a verbal measure would aid in adjudicating between these possibilities.

Reviewer: 2

Comments to the Author(s)

The question of whether motor span and the associated construct of kinaesthetic working memory (KWM) can be dissociated from visuospatial working memory (VSWM) as assessed by the Corsi Block task (which involves visual encoding and the generation of motor sequence to signal recall) is an important issue with potential relevance for the study of communicative movements such as co-speech gestures, signs in signed languages, and dance. The conclusion in Coulson and Wu (2014) that the two skills are dissociable was based on a study of 90 participants, a relatively low number for a psychometric study. Thus, replication of the observed low correlation coefficients (-0.01 with Corsi Span and 0.07 with Corsi Total) with more than double the original number of participants (viz. 200) would bolster this aspect of the original study. Likewise, the observation of reliable correlations between scores for movement span and Corsi

span would suggest the original result was a Type II error and would represent a genuine contribution to the literature.

The main way that the present study differs from the original is in the details of the Corsi Block Span test. In Wu and Coulson (2014), levels ranged from four to nine (c.f. in the present study, levels ranged from two to nine), and five trials (viz. sequences) were presented at each level (c.f. two trials were presented at each level in the present study). In Wu and Coulson (2014), participants only needed to correctly recall a single trial at a given level in order to advance to the next. In the present study, participants needed to correctly both trials at a given level in order to advance to the next level. In both studies, the total score was computed by summing all trials in which the participant correctly recalled all of the items in the sequence. However, the subtle differences between the administration of the test would predictably lead to differences in the total score values. Likewise, subtle differences between the way span scores were computed two studies might also lead to differences. In Wu and Coulson (2014), the span score was the highest level a participant completed (and because completion only required a single correct trial, there were no half points awarded). In the present study, participants were awarded a half-point to their span score if they advanced to a level and only had one correct trial. So, it is in principle possible that a failure to replicate might be attributable to these differences. However, given that there are a number of variants of the Corsi Block task that have been usefully deployed as a measure of visuospatial ability, these differences are unlikely to undermine the conclusions.

Moreover, another way that the present study differs from Wu and Coulson (2014), is that the authors here have employed two variants of the Corsi Block task – forward and backward. Although the two versions arguably assess slightly different cognitive abilities, with the backward span being more sensitive to the deployment of executive resources to track order (Cornoldi & Mammarella, 2008; Vandierendonck & Szmalec, 2003), they tend to be highly correlated with one another. The present study thus affords the possibility of conducting an even more robust assessment of the hypothesis that kinaesthetic and visuospatial WM abilities are dissociable from one another than the original study. This would involve both assessing the intercorrelation (or consistency via Cronbach's alpha) of each participant's forward and backward span scores in order to demonstrate their construct validity, and the separate correlation of each of the two Corsi span scores with participants' span scores on the motor working memory task.

Cornoldi, C., & Mammarella, I. C. (2008). A comparison of backward and forward spatial spans. *The Quarterly Journal of Experimental Psychology*, 61(5), 674-682.

Vandierendonck, A., & Szmalec, A. (2003). An asymmetry in the visuo-spatial demands of forward and backward recall in the Corsi blocks task. *Imagination, Cognition and Personality*, 23(2), 225-231.

Reviewer: 3

Comments to the Author(s)

In this replication study, the authors examine the validity of a recently published movement span task. Given the relatively limited sample size reported in the original study, I think the need for a replication is clear.

The method and results outlined here appear to be a faithful replication of the original paper. Some suggestions to improve the paper:

- I would like to see more details in the introduction on how the movement span task differs from previous attempts to determine the separability of movement from other WM subsystems (e.g., visuospatial WM). Why is the current approach showing a separate WM system while other papers have not? Similarly I'd like to see some further explanation on the rationale behind

looking at span scores and total accuracy. These are obviously related because span scores depend on accuracy. Why is it meaningful to look at both scores? I understand the need to replicate the approach from the original paper, but it wasn't clear to me why these are the best DVs.

- Also in the intro, the authors can do more to build the case for why a replication is needed. While I can infer that from the method, a dedicated section in the intro laying out the need would strengthen the rationale.

- Once the manuscript is complete with data and a discussion, I encourage the authors to discuss the limitations of the Corsi blocks task in measuring visuospatial WM and consider including at least one other visuospatial span task (e.g., symmetry or rotation span tasks) if data is not yet collected. Future work in this area should expand the visuospatial measures used to validate the movement span task to ensure it is actually separate from visuospatial WM.

Author's Response to Decision Letter for (RSOS-200237.R0)

See Appendix A.

Decision letter (RSOS-200237.R1)

08-Apr-2020

Dear Dr Nicoladis

On behalf of the Editor, I am pleased to inform you that your Manuscript RSOS-200237.R1 entitled "Toward a reliable measure of motor working memory: Revisiting Wu and Coulson's (2014) movement span task" has been accepted in principle for publication in Royal Society Open Science.

You may now progress to Stage 2 and complete the study as approved.

Please note that you must now register your approved protocol on the Open Science Framework (<https://osf.io/rr>), using the 'Submit your approved Registered Report' option and then the 'Registered Report Protocol Preregistration' option. Please use the Registered Report option even though your article is being accepted as a Stage 1 Replication. Further into the registration process, in the Journal Title field enter 'Royal Society Open Science (Replication article type, Results-Blind track)'. Please note that a time-stamped, independent registration of the protocol is mandatory under journal policy, and manuscripts that do not conform to this requirement cannot be considered at Stage 2. The protocol should be registered unchanged from its current approved state. Please include a URL to the protocol in your Stage 2 manuscript, and because you submitted via the Results-Blind track please note in the manuscript that the pre-registration was performed after data analysis (e.g. 'This article received results-blind in-principle acceptance (IPA) at Royal Society Open Science. Following IPA, the accepted Stage 1 version of the manuscript, not including results and discussion, was preregistered on the OSF (URL). This preregistration was performed after data analysis.')

We would be grateful if you could now update the journal office as to the anticipated completion date of your study.

Following completion of your study, we invite you to resubmit your paper for peer review as a Stage 2 Replication. Please note that your manuscript can still be rejected for publication at Stage 2 if the Editors consider any of the following conditions to be met:

- The Introduction and methods deviated from the approved Stage 1 submission (required).
- The authors' conclusions were not considered justified given the data.

We encourage you to read the complete guidelines for authors concerning Stage 2 submissions at: <https://royalsocietypublishing.org/rsos/replication-studies#AuthorsGuidance>. Please especially note the requirements for data sharing and that withdrawing your manuscript will result in publication of a Withdrawn Registration.

Once again, thank you for submitting your manuscript to Royal Society Open Science and I look forward to receiving your Stage 2 submission. If you have any questions at all, please do not hesitate to get in touch. We look forward to hearing from you shortly with the anticipated submission date for your stage two manuscript.

Kind regards,
Professor Chris Chambers
Royal Society Open Science
openscience@royalsociety.org

Author's Response to Decision Letter for (RSOS-200237.R1)

See Appendix B.

RSOS-200237.R2 (Revision)

Review form: Reviewer 1 (David Martinez)

Is the manuscript scientifically sound in its present form?

Yes

Is the language acceptable?

Yes

Do you have any ethical concerns with this paper?

No

Have you any concerns about statistical analyses in this paper?

No

Recommendation?

Accept with minor revision

Comments to the Author(s)

Interesting and much needed work! I have attached comments (Appendix C), most of which relate to some minor edits or my desire for a bit more elaboration.

Review form: Reviewer 3**Is the manuscript scientifically sound in its present form?**

Yes

Is the language acceptable?

Yes

Do you have any ethical concerns with this paper?

No

Have you any concerns about statistical analyses in this paper?

No

Recommendation?

Accept as is

Comments to the Author(s)

In reviewing this revised manuscript following the stage 1 review, the paper clearly meets the requirements for a well-executed replication study. The analysis and interpretation of the data relative to the original study are appropriate and reasonable. Specifically, I was pleased to see the current authors did not overstate the significance of their correlations relative to the original study given the larger sample size. A weak link here seems to make the most sense given the data. The authors also addressed the feedback from stage 2, which made for a stronger paper.

This is a valuable contribution and provides a beneficial replication to help guide future research using the movement span.

Decision letter (RSOS-200237.R2)

Dear Dr Nicoladis

On behalf of the Editor, I am pleased to inform you that your Stage 2 Replication submission RSOS-200237.R2 entitled "Toward a reliable measure of motor working memory: Revisiting Wu and Coulson's (2014) movement span task" has been accepted for publication in Royal Society Open Science subject to minor revision in accordance with the referee suggestions. Please find the referees' comments at the end of this email.

The reviewers and Subject Editor have recommended publication, but also suggest some minor revisions to your manuscript. Therefore, I invite you to respond to the comments and revise your manuscript.

Please also ensure that all the below editorial sections are included where appropriate (a non-exhaustive example is included in an attachment):

- Ethics statement

- Data accessibility

If you wish to submit your supporting data or code to Dryad (<http://datadryad.org/>), or modify your current submission to dryad, please use the following link:
<http://datadryad.org/submit?journalID=RSOS&manu=RSOS-200237.R2>

- Competing interests

- Authors' contributions

- Acknowledgements

- Funding statement

Because the schedule for publication is very tight, it is a condition of publication that you submit the revised version of your manuscript within 7 days (i.e. by the 08-May-2020). If you do not think you will be able to meet this date please let me know immediately.

- 1) A text file of the manuscript (tex, txt, rtf, docx or doc), references, tables (including captions) and figure captions. Do not upload a PDF as your "Main Document".
- 2) A separate electronic file of each figure (EPS or print-quality PDF preferred (either format should be produced directly from original creation package), or original software format)
- 3) Included a 100 word media summary of your paper when requested at submission. Please ensure you have entered correct contact details (email, institution and telephone) in your user account
- 4) Included the raw data to support the claims made in your paper. You can either include your data as electronic supplementary material or upload to a repository and include the relevant DOI within your manuscript
- 5) Included your supplementary files in a format you are happy with (no line numbers, Vancouver referencing, track changes removed etc) as these files will NOT be edited in production

Kind regards,
 Andrew Dunn
 Royal Society Open Science
openscience@royalsociety.org

on behalf of Professor Chris Chambers (Registered Reports Editor, Royal Society Open Science)
openscience@royalsociety.org

Associate Editor Comments to Author (Professor Chris Chambers):

Associate Editor: 1

Comments to the Author:

The Stage 2 manuscript was returned to two reviewers who assessed the Stage 1 submission. Happily, both are satisfied with the submission and judge that it meets the primary review criteria. Reviewer 2 is completely satisfied. Reviewer 1 offers some useful suggestions for minor revision, especially regarding the clarity of the results and issues to consider in the Discussion. The reviewer also suggests some changes the Introduction and Methods. For a regular paper this would be quite normal and routine, however for a Replication study at RSOS it is important that no unnecessary changes are made to the approved Stage 1 part of the manuscript, unless to correct typographical errors or clear errors of fact. Therefore, in revising, please do not make any changes to the approved Stage 1 part of the manuscript that do not fall into one of those categories.

Reviewers' comments to Author:

Reviewer: 1

Comments to the Author(s)

Interesting and much needed work! I have attached comments, most of which relate to some minor edits or my desire for a bit more elaboration.

Reviewer: 2

Comments to the Author(s)

In reviewing this revised manuscript following the stage 1 review, the paper clearly meets the requirements for a well-executed replication study. The analysis and interpretation of the data relative to the original study are appropriate and reasonable. Specifically, I was pleased to see the current authors did not overstate the significance of their correlations relative to the original study given the larger sample size. A weak link here seems to make the most sense given the data. The authors also addressed the feedback from stage 2, which made for a stronger paper.

This is a valuable contribution and provides a beneficial replication to help guide future research using the movement span.

Author's Response to Decision Letter for (RSOS-200237.R2)

See Appendix D.

Decision letter (RSOS-200237.R3)

Dear Dr Nicoladis:

It is a pleasure to accept your manuscript entitled "Toward a reliable measure of motor working memory: Revisiting Wu and Coulson's (2014) movement span task" in its current form for publication in Royal Society Open Science.

on behalf of Professor Chris Chambers (Subject Editor)
openscience@royalsociety.org

Appendix A

Dear Dr. Chambers,

We are grateful to the three reviewers and the associate editor for their careful reading of our manuscript and their constructive comments. We detail below exactly how we responded to their comments. Please continue to consider this manuscript for publication in Royal Society Open Science.

Yours sincerely,
Elena Nicoladis

Response to reviewers' comments

The original comments are in bold; our responses are not.

Associate Editor: 1

In relation to one point from Reviewer 3 (prompting the authors to "build the case for why a replication is needed") please note that this is not a criterion by which replication studies are assessed at RSOS, therefore the authors need not respond to this point in their revised manuscript.

Duly noted. Reviewer 2 gave us such great ideas for the justification that we could not resist adding a couple of sentences on further justification.

Reviewers' comments to Author:

Reviewer: 1

I believe the manuscript describes a valid and robust replication attempt however, I think we need a bit more detail so that readers can judge how close of a replication this is. On page 5 of the manuscript, lines 26-31, it is not clear what is meant by a "component." Moreover, an example of incorrect hand orientation is provided but, how accurate must one be? Is being off by 15 degrees acceptable? What about 45 degrees? I believe the instructions provided to raters and/or examples should be provided in an appendix.

We have corrected the word "component" to "parameter" and explained that we followed the catalogue of parameters laid out by Wu and Coulson (2014) in their supplementary materials.

It is also unclear whether participants were asked to recall body movements in serial order or any order---I assume free recall but this wasn't explicitly stated.

We have now stated that points were awarded independently of the order in which participants produced the movements.

I have just one more suggestion for the current manuscript. Wu and Coulson (2014) also provided measures of immediate verbal memory. Although it is somewhat out of the scope of this study, it would be helpful to include a verbal measure if it was part of the battery of tasks given in the larger study. If you find that the Corsi task correlates with the movement span task, then it may be because of domain-specific processes and/or domain-general processes---including a verbal measure would aid in adjudicating between these possibilities.

We were planning on having the participants' scores on a forward digit span task, which we have now incorporated.

Reviewer: 2

The conclusion in Coulson and Wu (2014) that the two skills are dissociable was based on a study of 90 participants, a relatively low number for a psychometric study. Thus, replication of the observed low correlation coefficients (-0.01 with Corsi Span and 0.07 with Corsi Total) with more than double the original number of participants (viz. 200) would bolster this aspect of the original study. Likewise, the observation of reliable correlations between scores for movement span and Corsi span would suggest the original result was a Type II error and would represent a genuine contribution to the literature.

We have incorporated some of these points in the introduction to help bolster the argument for the necessity of replication.

The main way that the present study differs from the original is in the details of the Corsi Block Span test. In Wu and Coulson (2014), levels ranged from four to nine (c.f. in the present study, levels ranged from two to nine), and five trials (viz. sequences) were presented at each level (c.f. two trials were presented at each level in the present study). In Wu and Coulson (2014), participants only needed to correctly recall a single trial at a given level in order to advance to the next. In the present study, participants needed to correctly both trials at a given level in order to advance to the next level. In both studies, the total score was computed by summing all trials in which the participant correctly recalled all of the items in the sequence. However, the subtle differences between the administration of the test would predictably lead to differences in the total score values. Likewise, subtle differences between the way span scores were computed two studies might also lead to differences. In Wu and Coulson (2014), the span score was the highest level a participant completed (and because completion only required a single correct trial, there were no half points awarded). In the present study, participants were awarded a half-point to their span score if they advanced to a level and only had one correct trial. So, it is in principle possible that a failure to replicate might be attributable to these differences. However, given that there are a number of variants of the Corsi Block task that have been usefully deployed as a measure of visuospatial ability, these differences are unlikely to undermine the conclusions.

The reviewer is correct that we used a more stringent manner of coding both total and span. We therefore have acknowledged these differences explicitly in the methods and removed the prediction of a magnitude difference from our predictions, concentrating instead on the pattern of intercorrelations.

Moreover, another way that the present study differs from Wu and Coulson (2014), is that the authors here have employed two variants of the Corsi Block task – forward and backward. Although the two versions arguably assess

slightly different cognitive abilities, with the backward span being more sensitive to the deployment of executive resources to track order (Cornoldi & Mammarella, 2008; Vandierendonck & Szmalec, 2003), they tend to be highly correlated with one another. The present study thus affords the possibility of conducting an even more robust assessment of the hypothesis that kinaesthetic and visuospatial WM abilities are dissociable from one another than the original study. This would involve both assessing the intercorrelation (or consistency via Cronbach's alpha) of each participant's forward and backward span scores in order to demonstrate their construct validity, and the separate correlation of each of the two Corsi span scores with participants' span scores on the motor working memory task.

We have now incorporated these references in our manuscript and will provide the intercorrelations between the tasks (see Table 2).

Reviewer: 3

Some suggestions to improve the paper:

- I would like to see more details in the introduction on how the movement span task differs from previous attempts to determine the separability of movement from other WM subsystems (e.g., visuospatial WM). Why is the current approach showing a separate WM system while other papers have not?

In this version of the manuscript, we have edited the second paragraph of the text so that it should be clear that many studies have argued for the separability of motor and visuospatial working memory (and many have argued for motor working memory to be a subset of visuospatial working memory). Studies on both sides of this argument have used similar methodology (e.g., interference tasks, correlations). We will include in the discussion that it would be interesting to see the field move toward thinking about how visual-spatial-motor memory are related and less about the either/or argument.

Similarly I'd like to see some further explanation on the rationale behind looking at span scores and total accuracy. These are obviously related because span scores depend on accuracy. Why is it meaningful to look at both scores? I understand the need to replicate the approach from the original paper, but it wasn't clear to me why these are the best DVs.

We have now added their rationale in the third paragraph, namely that they did so to make the movement span task analogous with other working memory tasks. We agree with the reviewer that the span and total are theoretically and empirically dependent on each other (as with the Corsi block task, by the way!). We will touch on this point in the discussion.

- Also in the intro, the authors can do more to build the case for why a replication is needed. While I can infer that from the method, a dedicated section in the intro laying out the need would strengthen the rationale.

We did so in the introduction by incorporating several points from Reviewer 2.

- Once the manuscript is complete with data and a discussion, I encourage the authors to discuss the limitations of the Corsi blocks task in measuring visuospatial WM and consider including at least one other visuospatial span task (e.g., symmetry or rotation span tasks) if data is not yet collected. Future work in this area should expand the visuospatial measures used to validate the movement span task to ensure it is actually separate from visuospatial WM.

We will add in the discussion that future research could include further measures of visuospatial memory in order to further test the claims about the relationship between visuospatial and motor memory. We agree that the field might overrely on Corsi blocks.

Appendix B

Dear Dr. Chambers,

We are pleased to submit the manuscript with the data as Stage 2 replication.

Yours sincerely,

Elena Nicoladis, PhD

Appendix C

Page 2, line 18: Might a better way to phrase the purpose be, “The purpose of the present study was to attempt to replicate Wu and Coulson’s (2014) observations of 1) a high correlation...2) a lack of a correlation between...”

Page 2, line 25, perhaps mention that you have a larger sample size here.

Page 2, line 27: I know “movement span and movement recall” are mentioned above, but using “movement span and movement recall” again would make it easier to follow---currently, it says “movement span and recall”.

Page 2, lines 25-27, you state that you found a high correlation for movement span and visuospatial wm capacity but you did not which makes the next sentence about significance confusing.

Page 3, line 3, I would say working memory is a *system* with the capacity to store unrelated units of information for a short period of time.

Working memory capacity is the individual differences variable.

Page 3-4, lines 36 -20 (on the next page), I’m not sure I understand the purpose of this paragraph. Is it to point out that there is evidence that memory for body movements is related to motor learning or is it to point out that some believe that memory body movements is independent from memory for other visuospatial information? And why are we being introduced to procedural memory here?

I think all of this information is important but there are several ideas contained in this one paragraph that should be separated. Perhaps introduce working memory for body movements and distinguish it from procedural memory (a very good point by the way) and maybe include evidence that memory for body movements also relates to learning motor tasks or begin a new paragraph with that info. Then another paragraph highlighting the argument that WM for body movements might be independent from visuospatial WM or it might not be.

Also, the last line, “If there were an independent motor working memory, it should be related to learning and processing motor information, like co-speech gestures.” But, isn’t it possible that a “dependent” motor working memory would be related to learning and processing motor information too?

Page 5, line 55, this would be a good place to explain why the verbal task was included. I believe I mentioned something in the last review...essentially for discriminant validity---you would not expect a verbal measure to correlate highly with the visuospatial and movement tasks but if all of the measures correlate fairly well (to my mind, perhaps $r_s > .3$), then perhaps it is due to a domain-general ability like attention (or Randy Engle’s executive attention).

Page 7, lines 28-54, I appreciate the extra detail provided about scoring and free recall!

Page 8, line 22, repeated → reported

Page 11, line 45 onward, the paragraph starts with verbal memory but then switches to visuospatial memory. Which is the focus of this paragraph?

Page 12, line 10, “in addition to we included...” perhaps drop the “in addition to” or follow that with “forward Corsi”.

Page 12, line 19, I would appreciate a bit more discussion about why visuospatial memory would be related to memory for movements. I don't want to speak for the authors, but it seems to me that one would expect the two to be related because, in sighted individuals, visuospatial memory can be supported by motor movements (e.g., using eye gaze or hand movements to repeat the Corsi pattern in a similar way as individuals will repeat letters during letter span) and memory for movements can be supported by visuospatial memory (e.g., remembering a body configuration and using that to recall the correct movement).

It would also be nice to read your thoughts on why Martinez and Singleton (2018) observed a larger correlation between a Corsi task and movement span. Specifically, why did they observe a larger correlation than the present study and Wu and Coulson's? I'll offer my thoughts. One possibility is that Martinez and Singleton's (2018) methods may have increased the role of domain-general processes. They used partial-scoring (Conway et al., 2005) and pseudo-randomized trials in their memory tasks such that participants did not know how many items they were about to try to encode. The partial scoring method and the fact that participants saw all trials (not just up to their span) may have increased the role of interference mitigation from supraspan lists (e.g., if a person's span is 5 but they saw a list of 9 items, they still get credit for anything they can remember (in the correct order) even in the face of the extra interference from the 4 items above their span). Moreover, Cowan (2008) claims that a difference between WM and STM tasks is that WM tasks somehow impede rehearsal---the fact that participants didn't know how many items were present in a trial may have reduced their ability to use a strategy (e.g., try to remember only the last few items). On the other hand, the methods used by the present authors as well as by Wu and Coulson may *increase* the chance that individuals can use strategies. For example, an individual who knows that the next list will contain 5 items may be able to focus their attention on the last 3 items and not worry about the first two because they know they cannot remember 5 items anyway. Future studies should probably manipulate strategy use. Another possibility is that Martinez and Singleton's sample included community members and therefore there may have been more variance, allowing for larger correlations. Finally, the larger correlation could be random

Page 12, line 47 onward, happy to see the importance of experience being noted!

Appendix D

Dear Dr. Chambers,

We have responded verified that the requested parts are now in the manuscript. We have responded to the comments by Reviewer #1 as outlined below.

All the best,
Elena Nicoladis, PhD

Reviewer: 1

Following the Associate Editor's instructions, we only addressed the points from reviewer 1 that were about the Stage 2 points.

Page 12, line 19, I would appreciate a bit more discussion about why visuospatial memory would be related to memory for movements. I don't want to speak for the authors, but it seems to me that one would expect the two to be related because, in sighted individuals, visuospatial memory can be supported by motor movements (e.g., using eye gaze or hand movements to repeat the corsi pattern in a similar way as individuals will repeat letters during letter span) and memory for movements can be supported by visuospatial memory (e.g., remembering a body configuration and using that to recall the correct movement).

We have added a sentence saying that we think that motor tasks often require visuospatial memory but some visuospatial tasks may require little motor memory, thereby explaining the weak link between the two.

it would also be nice to read your thoughts on why Martinez and Singleton (2018) observed a larger correlation between a corsi task and movement span. Specifically, why did they observe a larger correlation than the present study and Wu and Coulson's? I'll offer my thoughts. One possibility is that Martinez and Singleton's (2018) methods may have increased the role of domain-general processes. They used partial-scoring (Conway et al., 2005) and pseudorandomized trials in their memory tasks such that participants did not know how many items they were about to try to encode. The partial scoring method and the fact that participants saw all trials (not just up to their span) may have increased the role of interference mitigation from supraspan lists (e.g., if a person's span is 5 but they saw a list of 9 items, they still get credit for anything they can remember (in the correct order) even in the face of the extra interference from the 4 items above their span). Moreover, Cowan (2008) claims that a difference between WM and STM tasks is that WM tasks somehow impede rehearsal--the fact that participants didn't know how many items were present in a trial may have reduced their ability to use a strategy (e.g., try to remember only the last few items). On the other hand, the methods used by the present authors as well as by Wu and Coulson may increase the chance that individuals can use strategies. For example, an individual who knows that the next list will contain 5 items may be able to focus their attention on the last 3 items and not worry about the first two because they know they cannot remember 5 items anyway. Future studies should probably manipulate strategy use. Another possibility is that Martinez and Singleton's sample included community members and therefore there may have been more variance, allowing for larger correlations. Finally, the larger correlation could be random

The reviewer has given much greater thought to this than we have. We are not entirely convinced that there is a phenomenon to be interpreted : the correlation of .39 between visuospatial memory and movement memory in their study could simply be within the variability that occurs when working with humans.

Nonetheless, we have added some hand-waving along the lines that the reviewer suggested in the discussion.